# Biocompatible Nanocomposites Based on Poly(styrene-*block*-isobutylene-*block*-styrene) and Carbon Nanotubes for Biomedical Application

**DOI:** 10.3390/polym12092158

**Published:** 2020-09-22

**Authors:** Maria A. Rezvova, Arseniy E. Yuzhalin, Tatiana V. Glushkova, Miraslau I. Makarevich, Pavel A. Nikishau, Sergei V. Kostjuk, Kirill Yu. Klyshnikov, Vera G. Matveeva, Mariam Yu. Khanova, Evgeny A. Ovcharenko

**Affiliations:** 1Research Institute for Complex Issues of Cardiovascular Diseases, 650002 Kemerovo, Russia; yuzhalin@gmail.com (A.E.Y.); bio.tvg@mail.ru (T.V.G.); klyshnikovk@gmail.com (K.Y.K.); matveeva_vg@mail.ru (V.G.M.); khanovam@gmail.com (M.Y.K.); ov.eugene@gmail.com (E.A.O.); 2Research Institute for Physical Chemical Problems of the Belarusian State University, 220030 Minsk, Belarus; miraslau.makarevich@gmail.com (M.I.M.); nikishau@bsu.by (P.A.N.); kostjuks@bsu.by (S.V.K.); 3Faculty of Chemistry, Belarusian State University, 220006 Minsk, Belarus; 4Institute of Regenerative Medicine, Federal State Autonomous Educational Institution of Higher Education I.M. Sechenov First Moscow State Medical University of the Ministry of Health of the Russian Federation (Sechenov University), 119991 Moscow, Russia

**Keywords:** nanocomposites, carbon nanotubes, biocompatible materials, electrical conductivity, triblock copolymers

## Abstract

In this study, we incorporated carbon nanotubes (CNTs) into poly(styrene-*block*-isobutylene-*block*-styrene) (SIBS) to investigate the physical characteristics of the resulting nanocomposite and its cytotoxicity to endothelial cells. CNTs were dispersed in chloroform using sonication following the addition of a SIBS solution at different ratios. The resultant nanocomposite films were analyzed by X-ray microtomography, optical and scanning electron microscopy; tensile strength was examined by uniaxial tension testing; hydrophobicity was evaluated using a sessile drop technique; for cytotoxicity analysis, human umbilical vein endothelial cells were cultured on SIBS–CNTs for 3 days. We observed an uneven distribution of CNTs in the polymer matrix with sporadic bundles of interwoven nanotubes. Increasing the CNT content from 0 wt% to 8 wt% led to an increase in the tensile strength of SIBS films from 4.69 to 16.48 MPa. The engineering normal strain significantly decreased in 1 wt% SIBS–CNT films in comparison with the unmodified samples, whereas a further increase in the CNT content did not significantly affect this parameter. The incorporation of CNT into the SIBS matrix resulted in increased hydrophilicity, whereas no cytotoxicity towards endothelial cells was noted. We suggest that SIBS–CNT may become a promising material for the manufacture of implantable devices, such as cardiovascular patches or cusps of the polymer heart valve.

## 1. Introduction

Cardiovascular disease (CVD) is the most common cause of death globally [1]. Medical devices manufactured from polymer materials are increasingly used in the clinic to replace dysfunctional tissues and organs [2,3]. Despite their generally high functionality and biocompatibility, such materials frequently require additional modification [4]; for example, the incorporation of nanoparticles can substantially change the structure of a polymer and therefore improve its properties [5,6]. Carbon-based nanomaterials are currently actively tested for their potential usefulness in the modification of implantable medical devices [7,8], owing to their extraordinary high tensile strength which is attributed to the special organization of carbon atoms in a two-dimensional hexagonal grid [9]. Such cellular structure is characteristic of graphene and other allotropic modifications of carbon with sp^2^ hybridization such as fullerene and carbon nanotubes (CNTs).

The incorporation of carbon-based nanomaterials into polymer structure greatly enhances its mechanical properties, specifically tensile strength [10,11], which is of the upmost importance for implantable medical devices used in CVD, including vascular substitutes, chords, patches, and heart valves. On the other hand, the presence of carbon-based nanomaterials may lead to the reduced elasticity and increased rigidity of the final device, thus adversely affecting its function [12,13]. However, CNTs representing the coiled layers of graphene can be a promising material for the manufacture of implantable devices, owing to their flexibility and high tensile strength of about 50–150 GPa [14]. In addition to enhanced tensile strength, carbon-based nanomaterials may endow polymers with antimicrobial activity, electrical conductivity, or even improved hemocompatibility [5,15]. All of the above could be beneficial for implantable medical devices prone to thrombogenicity and infections.

Here, we synthesized poly(styrene-*block*-isobutylene-*block*-styrene) (SIBS), a biomaterial demonstrating high biocompatibility [16] yet limited tensile strength [17], to further modify its structure with CNTs at different concentrations. The resulting nanocomposite films were subjected to a uniaxial tension test followed by analyses of hydrophobicity, electric conductivity, and cytotoxicity. We reported that CNT-modified SIBS films demonstrate an improved tensile strength and electrical conductivity without being cytotoxic to endothelial cells in vitro. Thus, SIBS–CNT may become a promising material for medical device manufacture.

## 2. Materials and Methods

### 2.1. Synthesis of Poly(styrene-block-isobutylene-block-styrene)

Stabilized styrene (Sigma, St. Louis, MO, USA, >99%), methylene chloride (Ecos-1, Moscow, Russia, >99%), methylcyclohexane (Merck, Darmstadt, Germany, >99%), titanium (IV) chloride (Sigma, St. Louis, MO, USA, 99.9%) and 2, 6-dimethylpyridine (Acros, Geel, Belgium, 99%) were additionally purified by distillation. Isobutylene (Sigma, St. Louis, MO, USA, >99%) was dried by passing through a Drierite^™^ (W. A. Hammond drierite company, ltd, Xenia, OH, USA) drying system. Dicumyl chloride was prepared according to [18] by the hydrochlorination of dicumyl alcohol (Sigma, St. Louis, MO, USA, 97%). The purity of the final product was verified by ^1^H NMR spectroscopy (>99%).

The polymerization was conducted under inert argon according to [17]. Then, 38.6 mg dicumyl chloride (initiator, 0.167 mmol) was dissolved in 107 mL methylcyclohexane/methylene chloride mixture (3:2) followed by the addition of 0.161 mL (1.39 mmol) 2,6-dimethylpyridin. The system was cooled to −60 °C using an alcohol bath and 10.2 mL of isobutylene (119 mmol) pre-cooled to −60 °C was added. The temperature was further reduced to −80 °C, then 0.725 mL (6.6 mmol) of titanium (IV) chloride was added to start the polymerization. Fifty-seven minutes from the reaction start time, 16.0 mL of 2 M styrene solution in a methylcyclohexane/methylene chloride mixture (3:2) pre-cooled to −80 °C was added (32 mmol of styrene). To complete the reaction, 2 mL of pre-cooled ethanol was added 130 min from the start of the process. The resulting polymer was precipitated twice in a 10-fold excess of ethanol. The precipitate was separated by centrifugation, washed with ice-cold ethanol and vacuum-dried to constant weight.

The value of the average molecular weight and polydispersity of the resultant SIBS was determined by gel permeation chromatography using an Ultimate 3000 instrument (Thermo Fisher Scientific Dionex, Sunnyvale, CA, USA) equipped with a PLgel pre-column (7.5 × 50 mm, particle size 5 μm, Agilent Technologies, Santa Clara, CA, USA) and PLgel MIXED-C column (7.5 × 300 mm, particle size 5 μm, Agilent Technologies, Santa Clara, CA, USA). Detection was performed by refractometric and spectrophotometric (255 nm) detectors. Tetrahydrofuran (LiChrosolv^®^, Merck, Darmstadt, Germany, >99.9%) was used as an eluent. The elution rate was 1 mL/min at 30 °C. The polymer’s Mn and Mw/Mn values were calculated from the elution curves using EasiCal polystyrene standards (Agilent Technologies, Santa Clara, CA, USA) with Mn in the range of 580–400,000 g/mol and Mw/Mn ≤ 1.05. The calculation was performed using the Chromeleon 7.0 software (Thermo Fisher Scientific Dionex, Sunnyvale, CA, USA). The ^1^H NMR spectra of the polymer solutions in CDCl_3_ (Euriso-top^®^, Gif sur Yvette, France) with a concentration of ~15 g/L were measured at 25 °C on a Bruker AC-500 instrument (Billerica, MA, USA) operating at a frequency of 500 MHz. The chemical composition of polymer samples was determined using attenuated total reflection Fourier transform infrared spectroscopy (ATR-FTIR) using a ΦT-801 instrument (Infralum, Novosibirsk, Russia) in a spectral range of 400–650 cm^−1^.

### 2.2. Preparation of Nanocomposite Films

TUBALL™ CNTs with a diameter of 1.6 ± 0.4 nm, length of >5 μm and basic substance of ≥80%, were purchased from OCSiAl (Novosibirsk, Russia). CNTs were dispersed in 5 mL chloroform (Sigma, >99.9%) using an automatic ultrasonic disintegrator UD–20 (Techpan, Poland) with an output power of 180 W, a frequency of 22 kHz, and an oscillation amplitude of 8–16 μm. CNTs were treated by ultrasound for a total of 45 min. To prevent overheating, the ultrasound was delivered in cycles of a 30 s pulse followed by 15 s pause. A 5% *w*/*v* solution of SIBS in chloroform (5 mL) was added to disperse the CNTs followed by another round of sonication for 45 min. The resultant polymer solution was cast and air-dried at room temperature for 24 h followed by air drying using an Emitech SC 7640 sputter coater (Quorum Technologies, Ashford, UK) at <2 × 10^−2^ mbar. The synthesis pipeline is summarized in Figure 1. The chemical composition of the resulting samples was determined using an ATR-FTIR.

### 2.3. Microscopy

A 40 μL drop of SIBS–CNT mix dispersed in chloroform was placed on a glass slide followed by air drying. The bright field imaging of glass slides was performed using AXIO Imager A1 microscope (Zeiss, Jena, Germany) at ×1000 magnification. Alternatively, the slides were Au/Pd sputter-coated (7 nm) using EM ACE200 vacuum coater (Leica, Wetzlar, Germany) and imaged using a scanning electron microscope S-3400N (Hitachi, Tokyo, Japan) under high vacuum at an accelerating voltage of 5.0 kV in the secondary electron mode.

### 2.4. X-ray Microtomography (Micro-CT)

To evaluate the distribution of CNTs within SIBS, we performed a micro-CT imaging of SIBS–CNT films using a custom-built CT setup with a micrometer resolution (voltage 80 kV; current 48 μA; film thickness 2.54 μm). The obtained grayscale CT scans were imported into Avizo Software (Thermo), and a representative sample area of 1000 μm height × 1000 μm length × 100 μm depth was cropped for each sample, followed by a 3D reconstruction of X-ray density distribution. To define the range of CT densities of SIBS, we also tested an unmodified SIBS sample (0 wt%).

### 2.5. Tensile Testing

Uniaxial tensile testing was performed in accordance with the ISO 37:2017 standard. SIBS–CNT nanocomposites were subjected to uniaxial tension testing using a Z-series universal testing machine (Zwick/Roell) equipped with a 50 N rated force sensor at 37 °C. Polymer films were cut using a custom dumbbell-shape knife with a width of 2 mm and length of 10 mm (n = 8 per group). The samples were subjected to one loading cycle with a constant speed of 50 mm/min until rupture. Ultimate tensile dtrength (MPa) was defined as a maximum load based on the initial samples’ cross-sectional area. Uniform elongation (%) was defined as an engineering strain (strain) at maximum load. Young’s modulus (MPa) was determined in the physiological range at the beginning of the engineering stress–strain curve.

### 2.6. Contact Angle Measurement

The contact angle measurement was conducted using a sessile drop technique. A drop of distilled water (15 μL) was distributed on the surface of intact or CNT-modified SIBS films. All tests were performed at room temperature using a custom-built goniometer. The resulting images were processed using the contact angle plugin of the ImageJ software. The analysis was repeated 8 times for each study group.

### 2.7. Conductivity Assessment

The electrical conductivity was examined using an automated setup based on a Model 236 digital multimeter (Keithley Instruments). The resistance measurement was performed at a constant current according to the four-point probe test [19]. Measurements were repeated 5 times.

### 2.8. Cytotoxicity Assessment

Sterile SIBS films, intact or CNT-modified, were immobilized on the bottom of 24-well plates using agarose (5 wells per group). Cultured human umbilical vein endothelial cells (HUVEC) were trypsinized, washed with PBS, resuspended in EGM-2 (Lonza) supplemented with 5% fetal bovine serum, and seeded at a density of 4 × 10^4^ cells per well followed by 3 day incubation under cell culture conditions. Cell viability was assessed by fluorescent microscopy following their staining by 10 mg/mL Hoechst 33,342 (Thermo) and 30 μg/mL ethidium bromide (Sigma). Stained cells were imaged using an epifluorescent inverted microscope Axio Observer Z1 (Zeiss) at ×10 and ×20 magnification. Five random fields of view were imaged per well. The proportion of viable cells (ethidium bromide negative) (% VC) was calculated using the following formula: % VC = number of viable cells × 100%/number of all visible cells.

### 2.9. In Vitro Oxidation Assay

For the in vitro accelerated oxidative degradation assay, 0.1 mm thick intact or CNT-modified SIBS films of 5 × 5 mm (N = 5) were immersed in a solution of 0.1 M CoCl_2_ in 20% H_2_O_2_ at 37 °C as described previously [20,21]. After 14 days of exposure, the samples were thoroughly washed with dH_2_O, and then air-dried. The sample surface was then analyzed using a scanning electron microscope, S-3400N (Hitachi), under high vacuum at an accelerating voltage of 5.0 kV in the secondary electron mode. The chemical composition of the sample’s surface before and after treatment was determined using ATR-FTIR.

### 2.10. Statistical Analysis

Statistical analysis was performed using GraphPad Prism 6.0 (GraphPad Software, San Diego, CA, USA). The normality of distribution was assessed by the Kolmogorov–Smirnov test. For normally distributed variables, ANOVA with a post-hoc test was used to determine the difference between groups. For non-normally distributed variables, the Kruskal–Wallis test was used to determine the difference between groups. The results were presented as either the mean and standard deviation or a median and interquartile range, where appropriate. Differences were considered statistically significant at *p* < 0.05.

## 3. Results

### 3.1. Synthesis of CNT-Modified SIBS Films and Analysis of Their Structure

Triblock copolymers of isobutylene with styrene (M_n_ = 50,000 g/mol, M_w_/M_n_ < 1.3, central polyisobutylene block M_n_ = 36,000 g/mol, M_w_/M_n_ < 1.2) were synthesized by the sequential controlled cationic polymerization of isobutylene and styrene, employing the dicumyl chloride/TiCl_4_/2,6-dimethylpyridine initiating system at −80 °C. The gel permeation chromatography (GPC) curve of polyisobutylene completely shifts to the high molecular weight region after the addition of the second monomer (styrene) into the polymerization mixture, thus confirming the successful formation of the desired triblock copolymer (Figure 2A). The ^1^H NMR spectra of SIBS (Figure 2B) exhibited well resolved signals corresponding to the phenyl protons (6.2–7.2 ppm, *b*) of the polystyrene block and methyl protons (1.0–1.2 ppm, *c*) of the polyisobutylene block. In addition to these signals, the spectra showed lower intensity resonances corresponding to the methyl protons of the first isobutylene unit attached to the initiator at 0.78 ppm (*d*) and terminal chloromethine protons at 4.2–4.5 ppm (*a*). The weight fraction of polystyrene in SIBS was calculated from the integral intensities of the corresponding signals and was 32%, which is very close to the weight fraction of styrene in the initial comonomers mixture (33%). The resultant copolymer was further used for CNT incorporation.

Ultrasound-dispersed CNTs in chloroform and polymer solution were stable under static conditions at room temperature for 2 weeks. The SIBS solutions of different CNT concentrations (1%, 2%, 4%, 6%, and 8%) were cast to obtain nanocomposite films, which exhibited a homogenous structure and had a thickness of 100 ± 5 μm.

An ATR-FTIR spectrum of the resultant films (Figure 3) matched with previously reported SIBS spectra [22]. An absorbance peak at 2952 cm^−1^ corresponded to the asymmetric stretching vibrations of aliphatic –CH_3_ groups of isobutylene, while peaks at 2915–2847 cm^−1^ were determined by symmetric stretching vibrations of aliphatic –CH_3_ groups. The 1463 cm^−1^ peak was characteristic of the bending vibrations of aliphatic –CH_2_ and –CH_3_ groups. Absorbance peaks at 1373 and 1227 cm^−1^ correspond to the –CH_3_ groups of aliphatic chains and stretching vibrations of C–C bonds in the carbon backbone, respectively. The 1009 cm^−1^ peak reflected the stretching vibrations of C–C bonds in the carbon backbone. An increase in the CNT content led to a gradual reduction in the intensity of these absorption peaks due to the scattering of the reflected energy. The wide peak at 1600 cm^−1^ could be attributed to C–C vibrations in CNTs [23].

When analyzing the structure of SIBS–CNT films by optical microscopy, we observed a relatively uniform distribution of particles within the polymer matrix; however, sporadic CNT inclusions of a significant size exceeding the nanoscale level were observed in all CNT-modified SIBS samples (Figure 4). In certain areas, CNTs formed fiber networks characterized as bundles of interwoven nanotubes. An increase in the CNT content led to a gradual reduction in the films’ optical transparency and an increase in the number of CNT agglomerates (Figure 4A–F). The scanning electron microscopy (SEM) analysis of the nanocomposites’ surface showed a minimal difference between the intact CNT-modified SIBS films (Figure 4G–L); specifically, the unmodified SIBS had a regular surface pattern, whereas the SIBS–CNT films exhibited irregular depressions and elevations.

The analysis of films by micro-CT showed that the weighted average basic density of the unmodified SIBS was 65 units (range: 42–86 grayscale units based on 95% pixels). The addition of CNTs led to a gradual increase in the weighted average density, with the most prominent increment of density at CNT concentrations of >4 wt% (Figure 4S). An increase in CNT content was associated with a higher standard deviation of density, indicating an uneven distribution of CNTs. Qualitatively, the 3D reconstruction of CNT distribution revealed two distinct forms of particles within SIBS–CNT films: (i) CNTs uniformly dispersed in SIBS and (ii) CNT agglomerates of >10 μm^3^ (Figure 4M–R). Notably, both (i) and (ii) were found in all SIBS–CNT films regardless of their CNT concentration.

### 3.2. Tensile Testing of CNT-Modified SIBS Films

We observed a non-linear correlation between nanocomposites’ mechanical properties and CNT content (Figure 5). All the CNT-modified SIBS films demonstrated a significantly higher ultimate tensile strength as compared to unmodified SIBS samples (*p* < 0.01) (Figure 5A). Increasing the CNT content from 0 wt% to 4 wt% led to a linear increase in the Young’s modulus of SIBS films (Figure 5B). Upon reaching the CNT concentration of 4 wt%, the Young’s modulus soared to 52.7 MPa, which is 12 times higher than that of unmodified SIBS samples. In samples with a CNT concentration of 6 wt% and 8%, an even more dramatic rise in the Young’s modulus was documented (Figure 5B). The uniform elongation demonstrated a sharp drop in 1 wt% SIBS–CNT films in comparison with unmodified samples (*p* < 0.0001) (Figure 5C), while a further increase in the CNT content did not significantly affect this parameter.

### 3.3. Hydrophobicity Assessment of CNT-Modified SIBS Films

The incorporation of CNTs into the SIBS matrix resulted in a gradual reduction of the wetting angle, suggesting increased hydrophilicity (Figure 6). Although SIBS films with a CNT concentration of 1 wt% and 2 wt% demonstrated a similar wetting angle to that of control SIBS samples (all above 90°), a further CNT increase to 4 wt%, 6 wt% and 8 wt% led to a progressive reduction of the wetting angle to 84.4° ± 2.3°, 63.9° ± 3.2° and 66.7° ± 5.2°, respectively. Since materials with a contact angle below 90° can be considered as hydrophilic [24], 4 wt%, 6 wt% and 8 wt% SIBS–CNT films were characterized as hydrophilic in comparison with hydrophobic control SIBS samples.

### 3.4. Electrical Conductivity Measurement of CNT-Modified SIBS Films

The high electrical conductivity of a material is a requirement for the development of heart muscle patches [25]. An increase in CNT content resulted in the elevated electrical conductivity of SIBS–CNT films from 0.07 S/cm at 1 wt% CNT to 8.55 S/cm at 8 wt% CNT (Figure 7). Considering the insulating properties of the unmodified SIBS (approximately 10^−14^ S/cm), it can be concluded that CNT incorporation led to a substantial (about 10^4^ S/cm) increase in the electrical conductivity of the nanocomposite.

### 3.5. Cytotoxicity of CNT-Modified SIBS Films

To evaluate the potential cytotoxicity of CNT-modified SIBS films, we immobilized them on cell culture plates to investigate the growth of HUVECs on their surface. Three days post seeding, the absolute number (per 1 mm^2^) of HUVECs cultured on SIBS films regardless of CNT incorporation was significantly reduced in comparison with those grown in lab plastic (*p* < 0.05) (Figure 8). This suggests a slower growth of HUVECs on SIBS films. Notably, the incorporation of CNTs into SIBS did not affect HUVEC growth (Figure 8A,B). The relative number of viable cells was nearly identical in all study groups except for the positive control where the cells were treated with pure CNTs (*p* < 0.05) (Figure 8C). Thus, these data suggest that CNT-modified SIBS films exhibit cytotoxicity comparable to that of culture plastic and can therefore be considered biocompatible.

### 3.6. In Vitro Oxidative Stability of CNT-Modified SIBS Films

To evaluate the stability of CNT-modified SIBS films, we performed an in vitro oxidation assay by submerging the samples in a solution of 0.1 M CoCl_2_ in 20% H_2_O_2_ for 14 days. We did not observe any evidence of corrosion or cracking over the course of the experiment, suggesting that both intact and CNT-modified SIBS are resistant to oxidative degradation (Figure 9A). However, the oxidative stress led in the formation of a new IR peak at 1724 cm^−1^, which is characteristic of stretching vibrations of C=O or –COOH bonds (Figure 9B).

## 4. Discussion

Despite significant advances in the development of new polymers, their mechanical characteristics remain suboptimal, thus limiting applicability for manufacturing implantable medical devices for CVD and other fields of medicine [2]. In this study, we investigated SIBS, a material with well recognized biocompatibility as well as clinically approved for the coating of coronary stents [26]. Under cyclic loading, however, the SIBS demonstrated an insufficient tensile strength which resulted in its irreversible deformation, thus making this material unsuitable for the manufacture of artificial heart valves [27].

Here, we modified SIBS films with CNTs, a material of unique physicochemical properties [9]. We hypothesized that the resulting nanocomposite films would combine the strength of CNTs and the elasticity of SIBS [11]. However, the major requirement for that would be a strong interfacial interaction between the CNTs and the polymer [28]. Since both CNTs and SIBS are chemically inert and mainly interact through weak van der Waals forces [29], their interaction is mainly determined by the contact area [30,31]. Due to their great flexibility and high surface energy, CNTs tend to aggregate in bundles that are difficult to dissociate [11]. The ultrasound treatment was reported to effectively dissociate CNT bundles [32,33,34], but only at low CNT concentrations [14,35,36], which determined the range of 1–8 wt% CNT used in this study.

In Figure 4, the SEM image of an unmodified polymer demonstrates a regular pattern of the surface, probably determined by a particular arrangement of polystyrene and polyisobutylene copolymer blocks [37]. Previous studies showed that SIBS demonstrate a typical block copolymer morphology with microphase separation. Such morphology is characterized by sporadic branches as well as wavy, worm-like, cylindrical and spherical structures [37]. The surface of resulting nanocomposites had an irregular relief with sporadic depressions and elevations, indicating a disrupted polymer block packaging and chaotic distribution of CNTs within the matrix structure.

Microscopy and micro-CT 3D reconstruction images suggested that a complete homogeneous dispersion of CNTs in the SIBS was not achieved due to the formation of entangled agglomerates (bundles), which were particularly frequent in samples with a CNT content of >4 wt%. Such bundles may deteriorate the nanofiller’s affinity with a polymer and act as stress concentrators, thus reducing the mechanical strength [29]. In addition, they may negatively affect the properties of nanocomposites by reducing the total contact area between CNTs and SIBS [38]. Since the tensile strength of nanocomposites changed nonlinearly upon increasing the CNT concentration, with a lesser increment in the range of 0 to 2 wt% CNT, it could be assumed that small amounts of nanofiller do not lead to a strong connection between the agglomerates distributed in the polymer. Thus, the CNT content of <2 wt% does not substantially affect the mechanical strength of SIBS (Figure 10). It is likely that the 100% dispersion of CNTs would result in a linear dependence between CNT concentration and tensile strength, as reported previously [39]. We observed a high variation in Young’s modulus for 8 wt% SIBS–CNT films, which was likely due to the uneven distribution and bundling of CNT within the SIBS matrix. This is consistent with our micro-CT results showing a wide range of X-ray densities for 8 wt% SIBS–CNT samples. Similar to our findings, Cao et al. reported that the high concentration of CNTs leads to bundle aggregation in nanocomposites [40]. Reduction in the uniform elongation in CNT-modified samples can be explained by the relatively small ductility of CNTs with a rupture fracture strain not exceeding 10% [41,42]. Increasing the ultimate tensile strength also can be attributed by the CNT’s reinforcement and the initial mechanical properties of CNTs. CNTs can absorb the fracture energy due to the interfacial deformation between the CNT and the polymer matrix, such as delamination or detachment [43].

In this study, the maximum strength indices were documented for nanocomposites of 6–8 wt% CNT, which were similar to heart valve cusps [44] or vessels [45] which make them more preferred for use as components of medical devices for CVD compared to low CNT concentrations. However, the requirements for maximum strength can differ depending on the area of product application, and therefore the nanocomposites with a CNT content of 1–4 wt% may be of interest for further research too. Hydrophilic materials have demonstrated a better hemocompatibility due to formation of a hydration layer and reducing protein adsorption [24,46,47]; however, the interaction of a foreign body with components of blood is a complex process, and it is therefore impossible to judge on device hemocompatibility based on hydrophilicity testing alone. In this study, the water contact angle of SIBS films reduced upon CNT incorporation (at 4 wt%, 6 wt%, and 8 wt%), suggesting higher hydrophilicity and potentially, improved hemocompatibility. The observed increase in hydrophilicity is due to the CNT-mediated alteration of topography and/or the chemical composition of films’ surface [48]. The FTIR analysis indicated a decrease in the intensity of spectral peaks corresponding to methyl groups (Figure 3), suggesting that the high concentration of CNTs at the films’ surface reduced the number of highly hydrophobic methyl groups of isobutylene. In the study by Shokraei et al. [49], a nanocomposite of multiwalled CNTs and polyurethane nanofibers showed a reduction of the contact angle by increasing the concentration of CNTs as a result of dominant carboxyl groups on the surface. Another study reported an increase in the hydrophobicity of nanocomposites based on a polymer blend of polyvinylidene fluoride–polyacrilonitrile with an increase in the CNT content, which the authors attributed to the higher roughness of the material’s surface [50]. We suggest that the surface roughness factor did not play a significant role in our experiments, given the insignificant change in film topography upon the incorporation of CNTs. A high variation of the contact angle observed for the 1 wt% SIBS–CNT films could be explained by the uneven distribution of CNTs in the copolymer, which was consistent with our micro-CT data (Figure 4).

Cytotoxicity is a key parameter defining the biomedical applications of materials. SIBS–CNT films were previously reported to maintain the growth of L-929 fibroblasts [51], and the incorporation of CNTs into the polyvinyl alcohol matrix stimulated the proliferation of osteoblast-like MG-63 cells [52]. Here, we reported no negative effects of both SIBS and CNTs on HUVECs, suggesting their potential applicability in biomedicine.

CNT-containing nanocomposites exhibit excellent electrical conductivity [53,54], making them suitable for the manufacture of heart patches for the treatment of conduction disorders caused by coronary heart disease [55]. Unlike normal myocardium, fibrotic tissue resulting from myocardial infarction has a dramatically reduced electrical conductivity, which leads to the malfunction of the cardiac muscle [56]. Multiple conductive biocompatible polymer nanocomposites containing various carbon nanomaterials including CNTs have been studied to date [57,58,59]. The high electrical conductivity of CNTs is determined by their electronic structure, characterized by a delocalized electronic band [60]. For example, hydrogels based on chitosan and CNTs display excitation conduction rates similar to that of native myocardium and can thus be used to repair various cardiovascular defects without the risk of cardiac arrhythmias [61]. The majority of polymers display the conductivity of 10^−16^–10^−12^ S/m, however, it can be substantially increased by CNT incorporation [55,62]. In our studies, a sharp rise in conductivity upon the addition of CNTs was associated with a several-fold increase in the number of conducting channels (Figure 8). The maximum electrical conductivity was observed for SIBS films with 8 wt% CNTs, implicating this concentration as optimal for heart patch development. Our conductivity findings are consistent with those obtained for other CNT-based nanocomposites on the basis of polydimethylsiloxane [63] and polystyrene [64].

The long-term stability of styrene and isobutylene copolymer is determined by (1) high oxidative–hydrolytic–enzymatic stability (i.e., chemical inertness) of alternating secondary and quaternary carbon atoms in the backbone chain of the copolymer; (2) the high inertness of primary carbon atoms of the side groups; and (3) the absence of groups susceptible to hydrolysis such as ester, ether, amide and urea [16]. The high oxidative stability of the SIBS was also determined by the absence of multiple bonds. Others reported a superior oxidative stability of SIBS upon submerging a sample of SIBS in boiling concentrated (65%) nitric acid for 30 min [65]. While other elastomers used for implant applications were destroyed, SIBS samples remained relatively unaffected [65]. In our studies, both SIBS and SIBS–CNT films exhibited a resistance to oxidative destruction as evidenced, as well as no visible signs of corrosion and cracking on films’ surface [66]. Nonetheless, the detailed ATR-FTIR analysis showed some degree of sample oxidation. During a nanocomposite casting, a more hydrophobic and oxidation-resistant polyisobutylene block should appear at the film–air interface; CNT incorporation may change the distribution of blocks at the film-air interface, thus exposing the oxidation-prone polystyrene blocks. Alternatively, the emergence of carboxyl groups may be due to the oxidation of CNTs rather than SIBS. Even though CNTs are resistant to the action of many aggressive substances, some authors reported a formation of carboxyl and carbonyl groups on the CNT surface when exposed to H_2_O_2_ at 70 °C for 4 days [67].

## 5. Conclusions

In this study, we synthesized SIBS–CNT nanocomposite films to test their potential usefulness for biomedical application. The incorporation of 8 wt% CNT into the SIBS matrix resulted in a significantly improved tensile strength and electrical conductivity, however, the high rigidity of this nanocomposite may limit its functionality. The most optimal results were documented for SIBS films with a CNT content of 4–6 wt%, which exhibited both high tensile strength and conductivity.

## Figures and Tables

**Figure 1 polymers-12-02158-f001:**
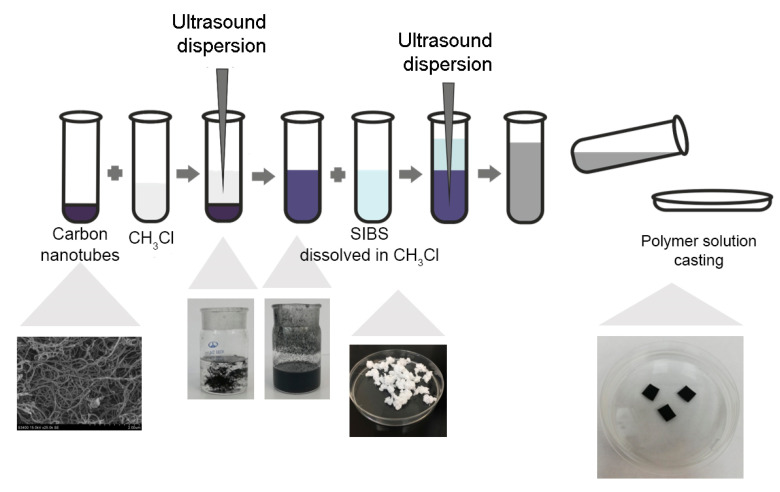
Study pipeline illustrating the fabrication of nanocomposite films based on carbon nanotubes incorporated into poly(styrene-*block*-isobutylene-*block*-styrene) (SIBS).

**Figure 2 polymers-12-02158-f002:**
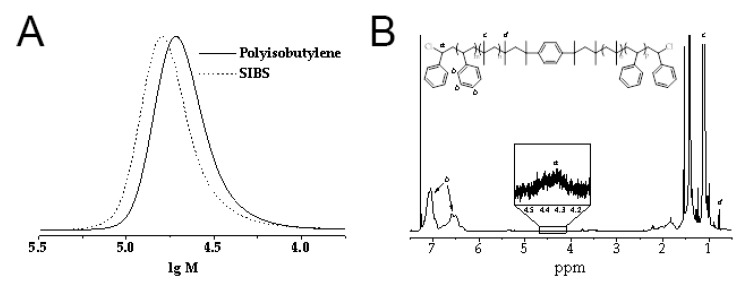
(**A**). Gel permeation chromatography (GPC) curves of the polyisobutylene middle block and the SIBS block-copolymer. (**B**). ^1^H NMR spectra of resultant copolymers.

**Figure 3 polymers-12-02158-f003:**
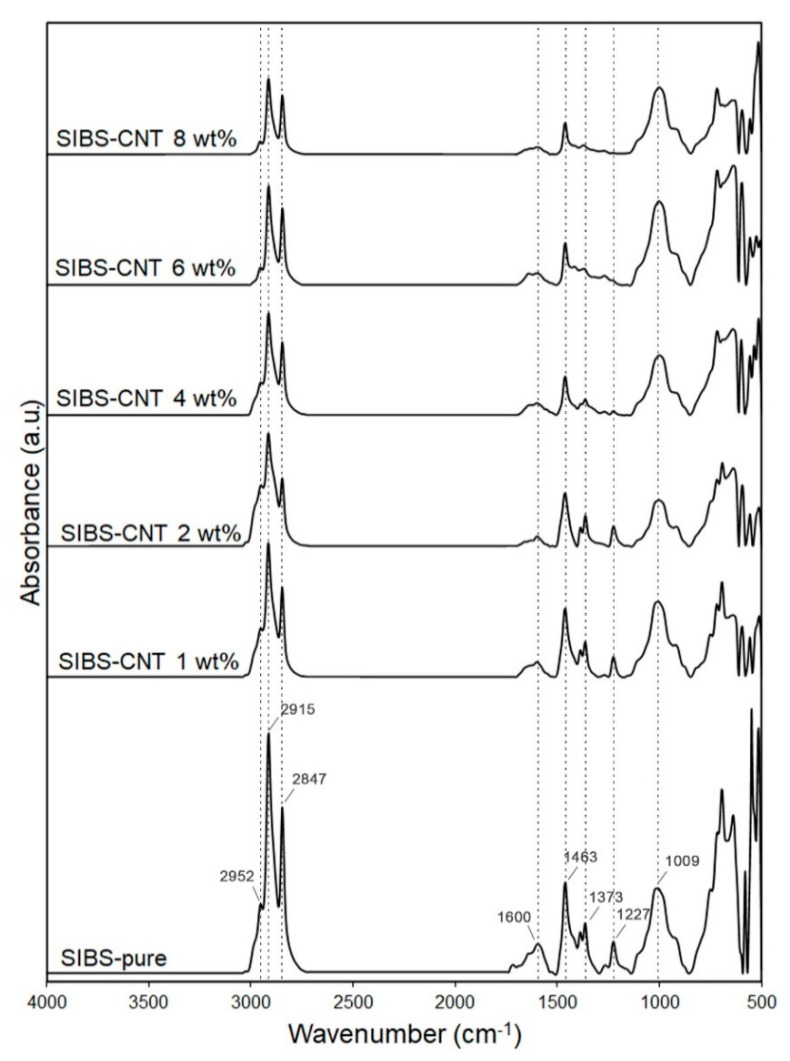
FTIR spectra of the intact and carbon nanotube (CNT)-modified SIBS.

**Figure 4 polymers-12-02158-f004:**
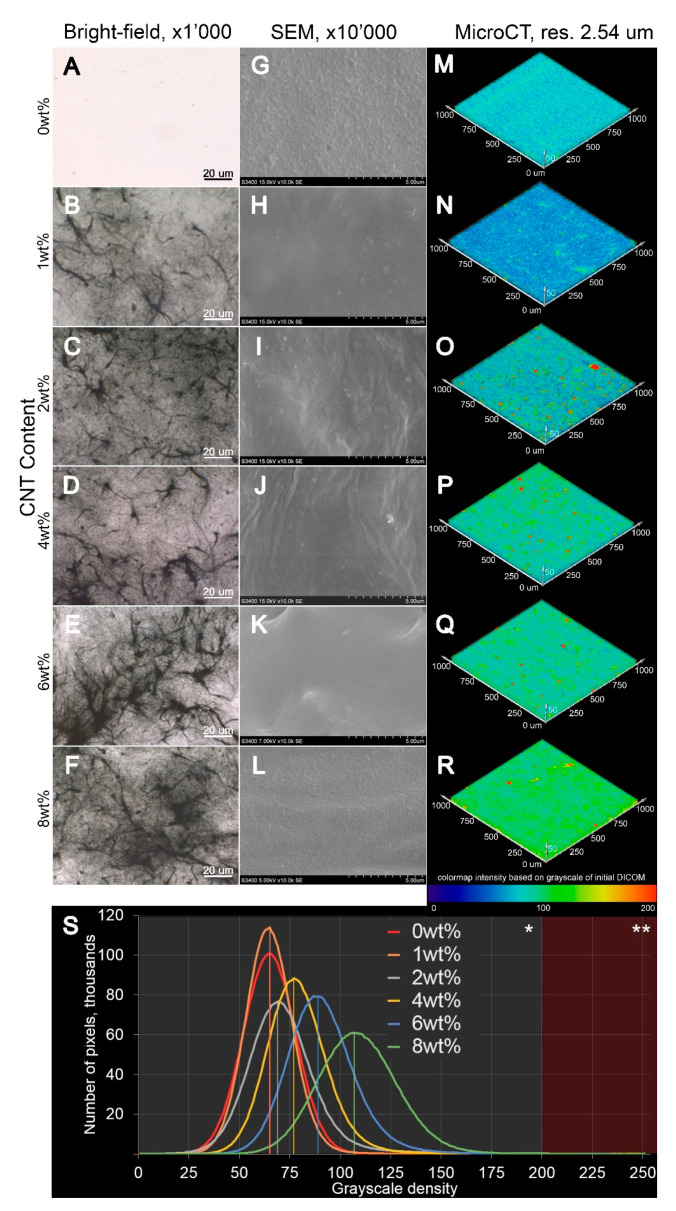
(**A**–**L**). Representative images of the intact and CNT-modified SIBS films acquired using optical (**A**–**F**), ×1000 magnification, and scanning electron microscopy (**G**–**L**), ×10,000 magnification. (**M**–**R**). The 3D reconstruction images of CNT distribution in SIBS films of 1000 μm height × 1000 μm length × 50 depth. For clarity, the visualization of 3D reconstructions was performed in the range of 0–200 units of X-ray density as the most informative, i.e., containing >99% of all pixels. The range of 201–255 units was excluded (contains <1% of all pixels). (**S**). The effect of CNT concentration on the X-ray density of SIBS films.

**Figure 5 polymers-12-02158-f005:**
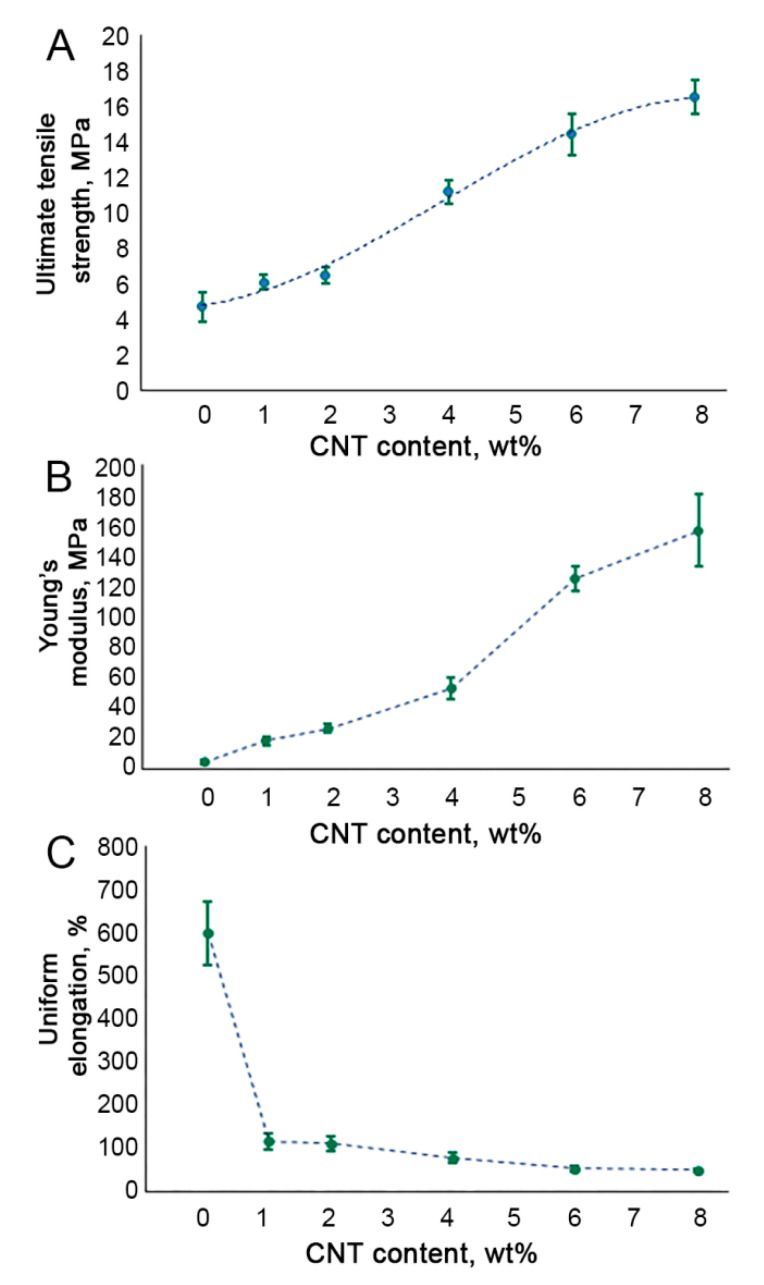
Tensile testing of SIBS–CNT films of different nanofiller concentrations: (**A**). the ultimate tensile strength; (**B**). Young’s modulus; (**C**). uniform elongation.

**Figure 6 polymers-12-02158-f006:**
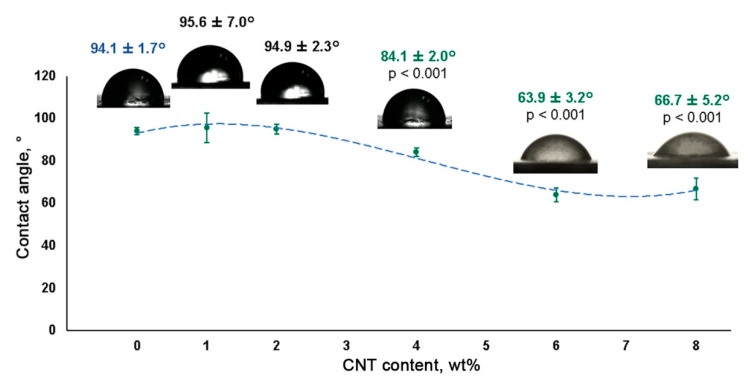
Water contact angles of SIBS–CNT films of different nanofiller concentrations.

**Figure 7 polymers-12-02158-f007:**
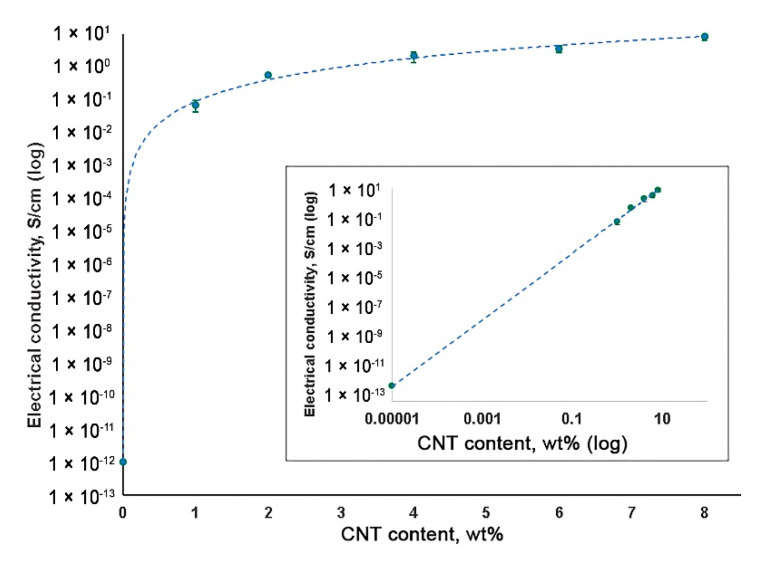
Electrical conductivity of SIBS–CNT films of different nanofiller concentrations.

**Figure 8 polymers-12-02158-f008:**
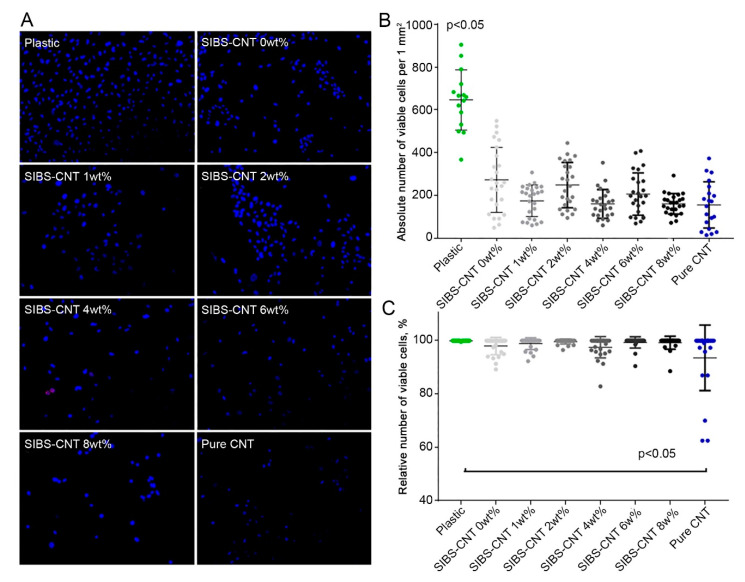
(**A**). Representative images of human umbilical vein endothelial cells (HUVEC) cultured on SIBS–CNT films for 3 days. Cells were stained with Hoechst (blue) and ethidium bromide (red). (**B**). The quantification of the cell number per mm^2^ of SIBS–CNT surface in the indicated groups. (**C**). Percentage of the viable cells in the indicated groups.

**Figure 9 polymers-12-02158-f009:**
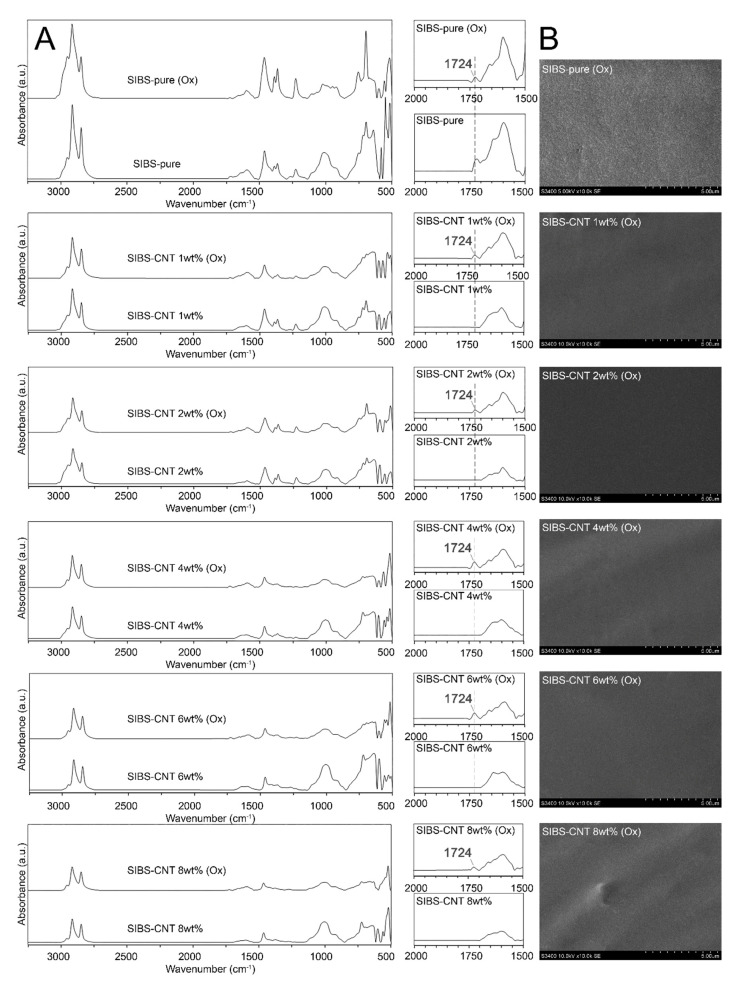
(**A**). SEM images of the surface of intact and CNT-modified SIBS films after 14 days of oxidation stress. (**B**). FTIR spectra of intact and CNT-modified SIBS films before and after 14 days of oxidation stress.

**Figure 10 polymers-12-02158-f010:**
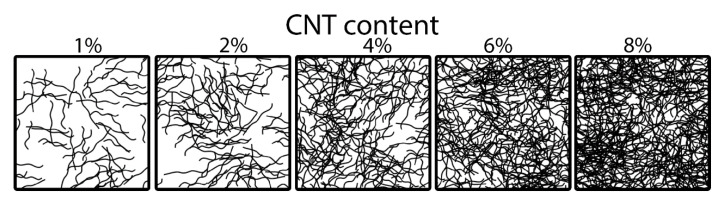
Schematic representation of CNT distribution in the SIBS matrices of different nanofiller concentrations.

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
