# Peer review of "Biocompatible Nanocomposites Based on Poly(styrene-block-isobutylene-block-styrene) and Carbon Nanotubes for Biomedical Application"

_polymers, 2020, doi:10.3390/polym12092158_

Round 1
Reviewer 1 Report
I would like to thank the authors for carefully and individually addressing all my questions/concerns/doubts. My feeling is that the manuscript may now be published in Polymers.
Reviewer 2 Report
In this revision, authors provided detailed data of their research. Now it's ready for publication after type setting. Furthermore, the schematic representation of CNT distribution in the SIBS (Fig. 10) is interesting but lack of evidence, computer simulation is suggested in future.
This manuscript is a resubmission of an earlier submission. The following is a list of the peer review reports and author responses from that submission.
Round 1
Reviewer 1 Report
This manuscript, titled “Biocompatible nanocomposites based on poly(styrene-block-isobutylene-block- styrene) and carbon nanotubes for biomedical application”, deals with the preparation of SIBS-CNT films and their characterization in terms of mechanical performance, hydrophobicity, electrical conductivity and cytotoxicity, as a preliminary study of the viability of these nanocomposites to be used as medical devices. Although the results and related discussion can be seen as preliminary, further characterization being required, my global feeling is that the results are interesting and relevant. Also, the way in which they are presented is well-structured. Nevertheless, there are still some concerns/questions/doubts that should be properly addressed by the authors before possible publication of the manuscript in Polymers:
Abstract
- Consider removing the first sentence of the Abstract: “Development of nanocomposites enables creating novel biomaterials with unique characteristics such as high tensile strength, biocompatibility, antibacterial activity, electrical conductivity etc.”. It is too generic; it would fit better in the Introduction section.
- What is the meaning of “We observed an uneven distribution of CNTs in the polymer matrix.”?
- CNT content: 0% to 8% - Why this range? Wt.% or vol.%?
- “led to a nonlinear increase in tensile strength of SIBS films.” – Please specify.
- “The relative elongation substantially decreased in 1% SIBS-CNT films …” – Is it significant?
- “We suggest that SIBS-CNT may become a promising material for medical device manufacture.” – What type of medical device?
Keywords
- Consider removing “Polymer hardening” (or alternatively replace it by a similar keyword)
Introduction
- What is the meaning of “… substantially improve polymer structure …”?
- “In this respect, CNTs, representing coiled layers of graphene, can be more promising owing to their flexibility and better ability to distribute in the polymer.” – Are carbon nanotubes really more flexible than graphene-based materials? And easier to distribute? A vast number of studies focus on the difficulties of properly distributing CNTs in a polymer matrix due to nanotube reaggregation.
Materials and methods
- “The polymerization was conducted under inert argon at according to …” – Replace by “The polymerization was conducted under inert argon according to …”
- “ml” instead of “mL”
- “2.2. Fabrication of nanocomposite films” – Better “2.2. Preparation of nanocomposite films”
- “2.5. Tensile testing” – Standard?
- “… determined in the range of small strains.” – Range?
- “2.7. Conductivity assessment” – Better “2.7. Electrical conductivity assessment”. What about sample preparation? Did you use conductive paint to enhance electrical contact during the measurements?
- “% VC = number of viable cells × 100% / number of all visible cells” – Consider embedding this in the text.
Results
- Figure 2 – Mention the different scales in the figure caption.
- “The scanning electron microscopy (SEM) analysis of nanocomposites’ surface showed no visible difference between SIBS-CNT films of various CNT concentrations and unmodified SIBS (Fig. 2G-L).” – There are some differences, especially between the non-reinforced film and those containing CNTs. Please explain.
- “Micro-CT 3D reconstruction …” – Further extend this paragraph; it would be interesting to comment on differences between the curves that appear in Figure 2M.
- Figure 3C – “Relative elongation”?
- Can you explain the high error bars of the 8% CNT film Young’s modulus and of the unreinforced film strain?
- “Young's modulus soared to 52.7” – Indicate the units (MPa).
- “The relative elongation demonstrated a sharp drop in 1% SIBS-CNT films” – Can you explain this sentence?
- Figure 4 – “Contact angle (º)” axis
- “Electrical conductivity of a material is required for development of heart muscle patches” – Could you extend on this?
- “Considering the dielectric properties of the …” – Replace by “Considering the insulating properties of the …”
- “… of HUVECs cultured on SIBS films regardless of CNT modification” – “Modification”?
Discussion
- However, SIBS had an insufficient tensile strength to be used as a material for artificial heart valves because of its irreversible deformation under a continuous load” – Rewrite
- “… however, the high stiffness of this composite should be considered.” - ?
My general feeling is that you could extend a bit more the Discussion section.
Conclusions
- “Additional modification of CNTs should be considered in order to improve their distribution in the SIBS matrix. Further in-depth studies of hemocompatibility and cyclic resistance are required to determine the applicability of SIBS-CNT nanocomposites for biomedicine.” – These seem more like further/future studies.
Reviewer 2 Report
CNT with block polymers such as SIS and SBS are widely investigated. The research is rough and the paper is written like experimental report. I suggest that the paper should be resubmitted after significant revision: Major: 1 FTIR and NMR characterizations are necessary data for your synthesized samples. 2 Biocompatibility is the first concern for biomaterials while biodegradability is also important for biopolymers. Long-term study is suggested to evaluate the degradability of your samples. 3 Many results are lack of discussions, such as, why the error of contact angle became larger when the content of CNT was 1%? 4 All mechanism of enhancement or improvement should be discussed. Minor: 5 Subheadings should be changed according to the template. 6 Reference should be listed with appropriate format. 7 Related references from Polymers could be cited such as: Chen, G.; Tang, W.; Wang, X.; Zhao, X.; Chen, C.; Zhu, Z. Applications of Hydrogels with Special Physical Properties in Biomedicine. Polymers 2019, 11, 1420.